# Quantum-Inspired Applications for Classification Problems

**DOI:** 10.3390/e25030404

**Published:** 2023-02-23

**Authors:** Cesarino Bertini, Roberto Leporini

**Affiliations:** 1Department of Management, University of Bergamo, via dei Caniana 2, I-24127 Bergamo, Italy; 2Department of Economics, University of Bergamo, via dei Caniana 2, I-24127 Bergamo, Italy

**Keywords:** quantum-inspired machine learning, classification, local approach

## Abstract

In the context of quantum-inspired machine learning, quantum state discrimination is a useful tool for classification problems. We implement a local approach combining the k-nearest neighbors algorithm with some quantum-inspired classifiers. We compare the performance with respect to well-known classifiers applied to benchmark datasets.

## 1. Introduction

Quantum-inspired machine learning is a new branch of machine learning based on the application of the mathematical formalism of quantum mechanics to devise novel algorithms. It has revealed how such algorithms have the potential to provide benefits in spite of lacking the computational power of quantum computers with several qubits. Some of these binary classifiers have been analyzed from a geometric perspective [1]. In this work, we implement some algorithms, based on quantum state discrimination, within a local approach in the feature space by taking into account elements close to the element to be classified. In particular, we perform multi-class classification directly (without using binary classifiers) based on Helstrom discrimination following an approach suggested by Blanzieri and Melgani [2], where an unlabeled data instance is classified by finding its *k* nearest training elements before running a support vector machine (SVM) over the *k* training elements. This local approach improves the accuracy in classification and motivates the integration with the quantum-inspired Helstrom classifier since the latter can be interpreted as a SVM with linear kernel [3]. It has the potential to offer comparable performance using less complexity because it uses few training points per test point.

The quantum-inspired classifiers require the encoding of the feature vectors into density operators and methods for estimating the distinguishability of quantum states like the Helstrom state discrimination and the pretty-good measurement (PGM). Quantum-inspired machine learning has revealed how relevant benefits for machine learning problems can be obtained using the quantum information theory even without employing quantum computers [4]. Moreover, as we will show below, our PGM within our algorithms is more efficient than the one proposed by these authors in the case of multiple preparations in the same state because it removes duplicates and null values in encoding. Quantum-inspired methods are used in applications that solve industry-relevant problems related to finance, optimization and chemistry [5,6,7,8,9].

In the experimental part, we present a comparison of the performances of the local quantum-inspired classifiers against well-known classical algorithms in order to show that the local approach can be a valuable tool for increasing the performances of this kind of classifier.

In Section 2, we review the notion of quantum encoding of data vectors into density operators and quantum-inspired classification based on quantum state discrimination [10,11,12,13]. In Section 3, we use the *k*-nearest neighbors algorithm (kNN) as a procedure to restrict the training set to the nearest elements around the test elements enabling the local execution of the quantum-inspired classifiers. In Section 4, we present and discuss some empirical results for evaluating the impact of locality in quantum-inspired classification comparing the performances of the proposed algorithms to classical methods over benchmark datasets. Furthermore, we compare quantum-inspired classifiers with SVMs within the local approach. In Section 5, there are the concluding remarks about the efficiency of local quantum-inspired classifiers.

## 2. Quantum-Inspired Classification

The first step of quantum-inspired classification is the *quantum encoding* that is any procedure to encode classical information into quantum states. In particular, we consider encoding of data vectors into density matrices on a Hilbert space H whose dimension depends on the dimension of the input space. *Density matrices* are positive semidefinite operators ρ such that trρ=1 and are the mathematical objects used to describe the physical states of quantum systems. *Pure states* are all the density matrices of the form $\rho =|\psi \rangle \langle \psi |$, with ‖ψ‖=1, which are the rank-1 projectors that can be directly identified with unit vectors up to a phase factor. Let ρ be a density operator on a *d*-dimensional Hilbert space $C^{d}$; it can be written in the following form:(1)$$\rho =\frac{1}{d}\left(I_{d}+\sqrt{\frac{d(d-1)}{2}}\sum_{j=1}^{d^{2}-1}b_{j}^{\left(\rho \right)}\sigma_{j}\right),$$
where ${\left\{\sigma_{j}\right\}}_{j=1,\ldots ,d^{2}-1}$ are the standard generators of the special unitary group SU(d), also called *generalized Pauli matrices*, and $I_{d}$ is the d×d identity matrix. The vector $b^{\left(\rho \right)}=\left(b_{1}^{\left(\rho \right)},\ldots ,b_{d^{2}-1}^{\left(\rho \right)}\right)$, with $b_{j}^{\left(\rho \right)}=\sqrt{\frac{d}{2(d-1)}}\mathrm{tr}\left(\rho \;\sigma_{j}\right)\in R$, is the *Bloch vector* associated with ρ which lies within the hypersphere of radius 1 in $R^{d^{2}-1}$. For d=2, the qubit case, the density matrices are in bijective correspondence to the points of the Bloch sphere in $R^{3}$, where the pure states are in one-to-one correspondence with the points of the spherical surface. For d>2, the points contained in the unit hypersphere of $R^{d^{2}-1}$ are not in bijective correspondence with density matrices on $C^{d}$, so the Bloch vectors do not form a ball but a complicated convex body. However, any vector within the sphere of radius $\frac{2}{d}$ gives rise to a density operator [14].

Complex vectors of dimension *n* can be encoded into density matrices of an (n+1)-dimensional Hilbert space H in the following way:(2)$$C^{n}∋x\mapsto |x\rangle =\frac{1}{\sqrt{{\left\|x\right\|}^{2}+1}}\left(\sum_{\alpha =0}^{n-1}x_{\alpha }|\alpha \rangle +|n\rangle \right)\in H,$$
where ${\left\{|\alpha \rangle \right\}}_{\alpha =0,\ldots ,n}$ is the computational basis of H, identified as the standard basis of $C^{n+1}$. The map defined in (Equation 2), called *amplitude encoding*, encodes x into the pure state $\rho_{x}=|x\rangle \langle x|$ where the additional component of $|x\rangle$ stores the norm of x. Nevertheless the quantum encoding $x\mapsto \rho_{x}$ can be realized in terms of the Bloch vectors $x\mapsto b^{\left(\rho_{x}\right)}$ saving space resources. The improvement of memory occupation within the Bloch representation is evident when we take multiple tensor products ρ⊗⋯⊗ρ of a density matrix ρ constructing a feature map to enlarge the dimension of the representation space [1].

Quantum-inspired classifiers are based on quantum encoding of data vectors into density matrices, calculations of centroids and various criteria of quantum state distinguishability such as: the Helstrom state discrimination, the pretty-good measurement [4,11] and the geometric construction of a minimum-error measurement [12]. Let us briefly recall the notion of quantum state discrimination. Given a set of arbitrary quantum states with respective a priori probabilities $R=\{\left(\rho_{1},p_{1}\right),\ldots ,\left(\rho_{N},p_{N}\right)\}$, in general there is no measurement process that discriminates the states without errors, i.e., a collection $E={\left\{E_{i}\right\}}_{i=1,\ldots ,N}$ of positive semidefinite operators such that $\sum_{i=1}^{N}E_{i}=I$, satisfying the following property: $\mathrm{tr}(E_{i}\rho_{j})=0$ when i≠j for all i,j=1,…,N. The probability of a successful state discrimination of the states in *R* performing the measurement *E* is:(3)$$P_{E}\left(R\right)=\sum_{i=1}^{N}p_{i}\mathrm{tr}\left(E_{i}\rho_{i}\right).$$
A complete characterization of the optimal measurement $E_{opt}$ that maximizes the probability (Equation 3) for $R=\{\left(\rho_{1},p_{1}\right),\left(\rho_{2},p_{2}\right)\}$ is due to Helstrom [10]. Let $\Lambda :=p_{1}\rho_{1}-p_{2}\rho_{2}$ be the *Helstrom observable* whose positive and negative eigenvalues are, respectively, collected in the sets $D_{+}$ and $D_{-}$. Consider the two orthogonal projectors:(4)$$P_{\pm }:=\sum_{\lambda \in D_{\pm }}P_{\lambda },$$
where $P_{\lambda }$ projects onto the eigenspace of λ. The measurement $E_{opt}:$$=\{P_{+},P_{-}\}$ maximizes the probability (Equation 3) that attains the *Helstrom bound*:(5)$$h_{b}\left(\rho_{1},\rho_{2}\right)=p_{1}\mathrm{tr}\left(P_{+}\rho_{1}\right)+p_{2}\mathrm{tr}\left(P_{-}\rho_{2}\right).$$
Helstrom quantum state discrimination can be used to implement a quantum-inspired binary classifier with promising performances. Let $\left\{\left(x_{1},y_{1}\right),\ldots ,\left(x_{M},y_{M}\right)\right\}$ be a training set with $x_{i}\in C^{n}$, $y_{i}\in \left\{1,2\right\}$ ∀i=1,…,M. Assume that, to encode the data points into quantum states by means of $C^{n}∋x\mapsto \rho_{x}\in S\left(H\right)$, one can construct the quantum centroids $\rho_{1}$ and $\rho_{2}$ of the two classes $C_{1,2}=\left\{x_{i}:y_{i}=1,2\right\}$:(6)$$\rho_{1,2}=\frac{1}{|C_{1,2}|}\sum_{x\in C_{1,2}}\rho_{x}$$
Let $\left\{P_{+},P_{-}\right\}$ be the Helstrom measurement defined by the set $R=\{\left(\rho_{1},p_{1}\right),\left(\rho_{2},p_{2}\right)\}$, where the probabilities attached to the centroids are $p_{1,2}=\frac{|C_{1,2}|}{|C_{1}\left|+\right|C_{2}|}$. The *Helstrom classifier* applies the optimal measurement for the discrimination of the two quantum centroids to assign the label *y* to a new data instance x, encoded into the state $\rho_{x}$, as follows:(7)$$y\left(x\right)=\left\{\begin{array}{cc} 1 & \mathrm{if}\;\mathrm{tr}\left(P_{+}\rho_{x}\right)\geq \mathrm{tr}\left(P_{-}\rho_{x}\right)\\ 2 & \mathrm{otherwise} \end{array}\right.$$
A strategy to increase the accuracy in classification is given by the construction of the tensor product of *q* copies of the quantum centroids $\rho_{1,2}^{⊗q}$ enlarging the Hilbert space where data are encoded. The corresponding Helstrom measurement is $\left\{P_{+}^{⊗q},P_{-}^{⊗q}\right\}$ and the Helstrom bound satisfies:(8)$$h_{b}\left(\rho_{1}^{⊗q},\rho_{2}^{⊗q}\right)\leq h_{b}\left(\rho_{1}^{⊗(q+1)},\rho_{2}^{⊗(q+1)}\right)\;\forall q\in N.$$
Increasing the dimension of the Hilbert space of the quantum encoding, one increases the Helstrom bound obtaining a more accurate classifier. The corresponding computational cost is evident; however, in the case of real input vectors, the space can be enlarged saving time and space by means of encoding into Bloch vectors.

Clearly, defining a quantum encoding is equivalent to selecting a feature map to represent feature vectors into a space of higher dimension. In the case of the considered quantum amplitude encoding $R^{2}∋\left(x_{1},x_{2}\right)\mapsto \rho_{\left(x_{1},x_{2}\right)}\in S\left(C^{3}\right)$, the nonlinear explicit injective function $\varphi :R^{2}\to R^{5}$ to encode data into Bloch vectors can be defined as follows:(9)$$\varphi \left(x_{1},x_{2}\right):=\frac{1}{x_{1}^{2}+x_{2}^{2}+1}\left(2x_{1}x_{2},2x_{1},2x_{2},x_{1}^{2}-x_{2}^{2},\frac{x_{1}^{2}+x_{2}^{2}-2}{\sqrt{3}}\right).$$
The mapped feature vectors are points on the surface of a hyper-hemisphere, with centroids of the classes, calculated as the means of these feature vectors, inside the hypersphere and can be rescaled to a Bloch vector as shown below.

In order to make the classification more accurate, one can increase the dimension of the representation space providing *k* copies of the quantum states, in terms of a tensor product, encoding data instances and centroids into density matrices $\rho^{⊗q}$. Bloch encoding allows an efficient implementation of feature maps; by removing null and repeated entries from the Bloch vector we obtain the following injective function for data encoding. Therefore, the Bloch representation allows an efficient storing of redundant elements of density matrices $\rho^{⊗q}$.

Let us consider a training set divided into the classes $C_{1},\ldots ,C_{M}$; assume we have any training point x encoded into the Bloch vector $b^{\left(x\right)}$ of a pure state on $C^{d}$. The calculation of the centroid of the class $C_{i}$, within this quantum encoding, must take into account that the mean of the Bloch vectors $b^{\left(i\right)}:=\frac{1}{|C_{i}|}\sum_{x\in C_{i}}b^{\left(x\right)}$ does not represent a density operator in general. In fact, for d>2 the points contained in the unit hypersphere of $R^{d^{2}-1}$ are not in bijective correspondence with density matrices on $C^{d}$. However, since any vector within the closed ball of radius $\frac{2}{d}$ gives rise to a density operator, a centroid can be defined in terms of a meaningful Bloch vector by a rescaling:(10)$${\hat{b}}^{\left(i\right)}:=\frac{2}{d|C_{i}|}\sum_{x\in C_{i}}b^{\left(x\right)}.$$

A method of quantum state discrimination for distinguishing more than two states $\left\{\left(\rho_{1},p_{1}\right),\ldots ,\left(\rho_{N},p_{N}\right)\right\}$ is the square-root measurement, also known as the *pretty-good measurement*, defined by:(11)$$E_{i}=p_{i}\rho^{-\frac{1}{2}}\rho_{i}\rho^{-\frac{1}{2}},$$
where $\rho =\sum_{i}p_{i}\rho_{i}$; PGM is the optimal minimum error when states satisfy certain symmetry properties [11]. Clearly, to distinguish between *n* centroids we need a measurement with at most *n* outcomes. It is sometimes optimal to avoid measurement and simply guess that the state is the a priori most likely state.

The optimal POVM ${\left\{E_{i}\right\}}_{i}$ for minimum-error state discrimination over
$$R=\{\left(\rho_{1},p_{1}\right),\ldots ,\left(\rho_{N},p_{N}\right)\}$$
satisfies the following necessary and sufficient Helstrom conditions [12]:(12)$$\Gamma -p_{i}\rho_{i}\geq 0\;\forall i=1,\cdots ,N,$$
where the Hermitian operator, also known as the *Lagrange operator*, is defined by $\Gamma :=\sum_{i}p_{i}\rho_{i}\;E_{i}$. It is also useful to consider the following properties which can be obtained from the above conditions:(13)$$E_{j}\left(p_{j}\rho_{j}-p_{i}\rho_{i}\right)E_{i}=0\;\forall i,j.$$
For each *i* the operator $\Gamma -p_{i}\rho_{i}$ can have two, one or no zero eigenvalues, corresponding to the zero operator, a rank-one operator, and a positive-definite operator, respectively. In the first case, we use the measurement $\left\{E_{i}=I,E_{i\neq j}=0\right\}$ for some *i* where $p_{i}\geq p_{j}$ ∀j, i.e., the state belongs to the a priori most likely class. In the second case, if $E_{i}\neq 0$, it is a weighted projector onto the corresponding eigenstate. In the latter case, it follows that $E_{i}=0$ for every optimal measurement.

Given the following Bloch representations:(14)$$\Gamma =\frac{1}{d}\left(a\;I_{d}+\sqrt{\frac{d(d-1)}{2}}\sum_{j=1}^{d^{2}-1}b_{j}^{}\sigma_{j}\right)\;\;,\;\rho_{i}=\frac{1}{d}\left(I_{d}+\sqrt{\frac{d(d-1)}{2}}\sum_{j=1}^{d^{2}-1}b_{j}^{\left(i\right)}\sigma_{j}\right)\;\;,$$
in order to determine the Lagrange operator in $C^{d}$ we need $d^{2}$ independent linear constraints:(15)$$2p_{i}\left(a-{\hat{b}}^{\left(i\right)}\cdot b-\frac{p_{i}}{2}\left(1-\right|{\hat{b}}^{\left(i\right)}{|}^{2})\right)=a^{2}-{\left|b\right|}^{2}.$$
A measurement with more than $d^{2}$ outcomes can always be decomposed as a probabilistic mixture of measurements with at most $d^{2}$ outcomes. Therefore, if the number of classes is greater than or equal to $d^{2}$ and we get $d^{2}$ linearly independent equations, we construct the Lagrange operator and derive the optimal measurements. From the geometric point of view, we obtain the unit vectors corresponding to the rank-1 projectors $E_{i}=\frac{1}{d}\left(I_{d}+\sqrt{\frac{d(d-1)}{2}}\sum_{j=1}^{d^{2}-1}n_{j}^{\left(i\right)}\sigma_{j}\right)$ where $n^{\left(i\right)}=\frac{{\hat{b}}^{\left(i\right)}-ab}{|{\hat{b}}^{\left(i\right)}-ab|}\in R^{d^{2}-1}$ giving the POVM of the measurement. It is also possible to further partition the classes in order to increase the number of centroids and of the corresponding equations. The classification is carried out in this way: an unlabeled point $\hat{x}$ is associated with the first label *y* such that $b^{\left(\hat{x}\right)}\cdot n^{\left(y\right)}=\max_{i}b^{\left(\hat{x}\right)}\cdot n^{\left(i\right)}$, where $d=\lceil \sqrt{length(x)+2}\rceil$.

## 3. Local Quantum-Inspired Classifiers

In the implementation, we consider the execution of the classifiers described above after a selection of the *k* training elements that are closest to a considered unclassified instance.

The *k*-nearest neighbors algorithm (kNN) is a simple classification algorithm which consists of the following steps:The computation of the chosen distance metric between the test element and the training elements;The extraction of the *k* elements closest to the test instance;The assignment of the class label through a majority voting based on the labels of the *k* nearest neighbors.

In the following, we apply the kNN for the extraction of the closest elements to the test element then the classification is performed by a quantum-inspired algorithm instead of majority voting. On the one hand, given a test element, the kNN can be executed over the data vectors in the input space, e.g., considering the Euclidean distance, then the *k* neighbors can be encoded into density matrices and used for a quantum-inspired classification. On the other hand, the entire dataset can be encoded into density matrices and the kNN selects the *k* neighbors evaluating an operator distance among quantum states. In the latter case, we consider the *Bures distance* that is a quantum generalization of the Fisher information and a distance derived by the super-fidelity. The Bures distance is defined by:(16)$$d_{B}\left(\rho_{1},\rho_{2}\right)=\sqrt{2\left(1-\sqrt{F(\rho_{1},\rho_{2})}\right)},$$
where the fidelity between density operators is given by $F\left(\rho_{1},\rho_{2}\right)={\left(\mathrm{tr}\sqrt{\sqrt{\rho_{1}}\rho_{2}\sqrt{\rho_{1}}}\right)}^{2}$. Let us note that the fidelity reduces to $F\left(\rho_{1},\rho_{2}\right)=\langle \psi_{1}|\rho_{2}|\psi_{1}\rangle$ when $\rho_{1}=|\psi_{1}\rangle \langle \psi_{1}|$. Therefore the Bures distance between the pure state $\rho_{1}$ and the arbitrary state $\rho_{2}$ can be expressed in term of the Bloch representation as follows:(17)$$d_{B}\left(\rho_{1},\rho_{2}\right)=\sqrt{2\left(1-\sqrt{\frac{1}{d}\left(1+\left(d-1\right)b^{\left(1\right)}\cdot b^{\left(2\right)}\right)}\right)}\equiv D_{B}\left(b^{\left(1\right)},b^{\left(2\right)}\right),$$
where $b^{\left(1\right)}$ and $b^{\left(2\right)}$ are the Bloch vectors of $\rho_{1}$ and $\rho_{2}$, respectively, and *d* is the dimension of the Hilbert space of the quantum encoding. The special form (Equation 17) of the Bures distance, expressed in terms of Bloch vectors, is relevant for our purpose because data vectors can be encoded into pure states and in general quantum centroids are mixed states.

An alternative distance can be defined via super-fidelity [15]
(18)$$d_{G}\left(\rho_{1},\rho_{2}\right)=\sqrt{1-G(\rho_{1},\rho_{2})},$$
where the super-fidelity between density operators is given by
$$G\left(\rho_{1},\rho_{2}\right)=\mathrm{tr}\rho_{1}\rho_{2}+\sqrt{\left(1-\mathrm{tr}\rho_{1}^{2}\right)\left(1-\mathrm{tr}\rho_{2}^{2}\right)}.$$
Notice that the super-fidelity reduces to $G\left(\rho_{1},\rho_{2}\right)=\langle \psi_{1}|\rho_{2}|\psi_{1}\rangle$ when $\rho_{1}=|\psi_{1}\rangle \langle \psi_{1}|$. This distance can be expressed in term of the Bloch representation as follows:(19)$$D_{G}\left(b^{\left(1\right)},b^{\left(2\right)}\right)=\sqrt{1-\frac{1}{d}\left(1+\left(d-1\right)\left(b^{\left(1\right)}\cdot b^{\left(2\right)}+\sqrt{\left(1-\right|b^{\left(1\right)}{|}^{2})(1-|b^{\left(2\right)}{|}^{2})}\right)\right)},$$
where $b^{\left(1\right)}$ and $b^{\left(2\right)}$ are the Bloch vectors of $\rho_{1}$ and $\rho_{2}$, respectively, and *d* is the dimension of the Hilbert space of the quantum encoding. The inner distance between the corresponding Bloch vectors represents the angle θ between the unit vectors $\left(b^{\left(1\right)},\sqrt{1-|b^{\left(1\right)}{|}^{2}}\right)$ and $\left(b^{\left(2\right)},\sqrt{1-|b^{\left(2\right)}{|}^{2}}\right)$, which is normalized to be 1:(20)$${\hat{D}}_{G}\left(b^{\left(1\right)},b^{\left(2\right)}\right)=\frac{\arccos\left(b^{\left(1\right)}\cdot b^{\left(2\right)}+\sqrt{\left(1-\right|b^{\left(1\right)}{|}^{2})(1-|b^{\left(2\right)}{|}^{2})}\right)}{\pi }.$$
For pure states the inner distance corresponds to the Fubini-Study distance.

In Algorithm 1, the locality is imposed by running the kNN on the input space finding the training vectors that are closest to the test element; then there is the quantum encoding into pure states and a quantum-inspired classifier (Helstrom, PGM or geometric Helstrom) is locally executed over the restricted training set. In Algorithm 2, the test element and all the training elements are encoded into Bloch vectors of pure states then a kNN is run w.r.t. the Bures distance to find the nearest neighbors in the space of the quantum representation; then a quantum-inspired classifier is executed with the training instances corresponding to the closest quantum states.
**Algorithm 1** Local quantum-inspired classification based on kNN in the input space before the quantum encoding. The distance can be: Euclidean, Manhattan, Chessboard, Canberra or Bray–Curtis.**Require:** Dataset *X* of labeled instances, unlabeled point $\hat{x}$
**Ensure:** Label of $\hat{x}$
     find the *k* nearest neighbors $x_{1},\ldots ,x_{k}$ to $\hat{x}$ in *X* w.r.t. the Euclidean distance
     encode $\hat{x}$ into a pure state $\rho_{\hat{x}}$
     **for** 
j=1,…,k 
**do**
         encode $x_{j}$ into a pure state $\rho_{x_{j}}$
     **end for**
     run the quantum-inspired classifier with training points encoded into ${\left\{\rho_{x_{j}}\right\}}_{j=1,\ldots ,k}$.


**Algorithm 2** Local quantum-inspired classification based on kNN in the Bloch representation after quantum encoding. The distance can be: Bures, Super-Fidelity or Inner.**Require:** Dataset *X* of labeled instances, unlabeled point $\hat{x}$
**Ensure:** Label of $\hat{x}$
     encode $\hat{x}$ into a Bloch vector $b^{\left(\hat{x}\right)}$ of a pure state
     **for** 
x∈X 
**do**
           encode x into a Bloch vector $b^{\left(x\right)}$ of a pure state
     **end for**
     find the *k* nearest neighbors to $b^{\left(\hat{x}\right)}$ in ${\left\{b^{\left(x\right)}\right\}}_{x\in X}$ w.r.t. the distance $D_{B}$
     run the quantum-inspired classifier over the *k* nearest neighbors.


A local quantum-inspired classifier can be defined without quantum state discrimination but considering a *nearest mean classification* such as the following: after the quantum encoding we perform a kNN selection and calculate the centroid of each class considering only the nearest neighbors to the test element, finally we assign the label according to the nearest centroid as schematized in Algorithm 3.
**Algorithm 3** Local quantum-inspired nearest mean classifier.**Require:** Training set *X* divided into *n* classes $C_{i}$, unlabeled point $\hat{x}$

**Ensure:** Label of $\hat{x}$

    encode $\hat{x}$ into a Bloch vector $b^{\left(\hat{x}\right)}$ of a pure state 
    **for** 
x∈X 
**do**

            encode x into a Bloch vector $b^{\left(x\right)}$ of a pure state 
    **end for**

    find the neighborhood $K=\{b^{\left(x_{1}\right)},\ldots ,b^{\left(x_{k}\right)}\}$ of $b^{\left(\hat{x}\right)}$ w.r.t. the distance $D_{B}$

    **for** 
i=1,…,n
**do**

            construct the centroid ${\hat{b}}^{\left(i\right)}=\frac{2}{d|C_{i}^{k}|}\sum_{x\in C_{i}^{k}}b^{\left(x\right)}$ where $C_{i}^{k}:=\left\{x\in C_{i}:b^{\left(x\right)}\in K\right\}$

    **end for**

    find the closest centroid ${\hat{b}}^{\left(l\right)}$ to $\frac{2}{d}b^{\left(\hat{x}\right)}$ w.r.t. the distance $D_{B}$

    **return** label of the class $C_{l}$


## 4. Results and Discussion

In this section, we present some numerical results obtained by the implementation of the local quantum-inspired classifiers with several distances compared to well-known classical algorithms. In particular, we consider the SVM with different kernels: linear, radial basis function and sigmoid. Then, we run a random forest, a naive Bayes classifier and the logistic regression. In order to compare the results with previous papers, we take into account the following benchmark datasets from the PMLB public repository [16]: analcatdata_aids, analcatdata_asbestos, analcatdata_bankruptcy, analcatdata_boxing1, analcatdata_cyyoung9302, analcatdata_dmft, analcatdata_happiness, analcatdata_japansolvent, analcatdata_lawsuit, appendicitis, biomed, breast_cancer, iris, labor, new_thyroid, phoneme, prnn_fglass, prnn_synth, tae and wine_recognition. For each dataset we randomly select 80% of the data to create a training set and use the residual 20% for the evaluation. We repeated the same procedure 10 times and calculated the average accuracy in Table 1. Certainly, it is possible to compare the performances based on different statistic indices including Matthews correlation coefficient, F-measure and Cohen’s parameter.

We observe that the performances of the local quantum-inspired classifiers turn out to be definitely more accurate, where the hyperparameter *k* is set equal to the number of classes in the dataset. This value is reasonable to construct the centroids of the classes. In particular, Algorithm 1 with the Euclidean distance is the most accurate classifier for the datasets analcatdata_boxing1, analcatdata_happiness, biomed, prnn_fglass and wine_recognition, while the Manhattan distance is best for analcatdata_aids, analcatdata_japansolvent, breast_cancer, iris and tae, the Chessboard distance is best for *analcatdata_cyyoung9302* and analcatdata_lawsuit, and the Bray–Curtis distance is best for analcatdata_bankruptcy and appendicitis. Algorithm 2 with the Bures distance outperforms Algorithm 1 and 3 for analcatdata_dmft and produces the same accuracy for labor. Algorithm 3 with the Bures distance is the most accurate classifier for analcatdata_asbestos, new_thyroid, phoneme and prnn_synth. Algorithm 1 uses a k-d tree in the training set, while the other two use a k-d tree in the corresponding Bloch vector space. The time complexity to construct the k-d tree is usually O(dnlogn), where *n* is the cardinality of the training set and *d* the length of each vector, while the space complexity is O(dn). The query to find the *k* nearest neighbors takes O(klogn). The time complexity of PGM is $O(cd^{3})$ and is O(dm) for the classification of the *m* elements of the test set in *c* classes. Our algorithm is more efficient than the one presented in [4] in the presence of multiple copies because it remove nulls and duplicates. In particular, we consider only 20 values instead of 81 matrix elements of $\rho_{\left(x_{1},x_{2}\right)}⊗\rho_{\left(x_{1},x_{2}\right)}$, 51 values instead of 729 for $\rho_{\left(x_{1},x_{2}\right)}⊗\rho_{\left(x_{1},x_{2}\right)}⊗\rho_{\left(x_{1},x_{2}\right)}$ and so on. In a future paper, we will analyze in detail the complexity of such algorithms in the average case and in the worst case. For instance, one can construct the ball tree for clustered data instead of the k-d tree and consider different search techniques.

In Table 2, we show the methods that provided the best accuracy, with the respective execution times, compared with the classical method. These experimental results are promising and show that the methods are efficient when run on classical computers. Algorithm 3 with the Bures distance is not efficient for phoneme, but Algorithm 1 with the Euclidean distance is: 1.951 s with an average accuracy of 0.897. We will study in a future work how to also apply the local methods in implementations on quantum computers.

Let us focus on multi-class datasets for the comparison with the kNNSVM method proposed by Blanzieri and Melgani [2]. This method requires the choice of the hyperparameter *k*, and as is well known from the standard kNN algorithm, there is no general strategy to choose *k* a priori. In Table 3, the results obtained for some *k* values of the kNNSVM are shown. For *analcatdata_dmft*, kNNSVM presents an average accuracy that is only 2% lower than Algorithm 2 but requires 17 elements per test element instead of 6. For *analcatdata_happiness*, kNNSVM yields an average accuracy that is 10% lower than Algorithm 1 and requires 14 elements per test element instead of 3. However, kNNSVM outperforms local quantum-inspired classifiers for *iris* and *tae*, but only for the latter requires fewer elements, while for *wine_recognition* they are comparable. For *new_thyroid and prnn_fglass*, the best results are obtained with the nearest neighbor method, but with lower accuracy than Algorithms 1 and 3, respectively.

## 5. Conclusions

The present paper focuses on the implementation of classification algorithms based on quantum state discrimination. A novel contribution is the local approach adopted to execute the classifier, not over the entire training set, but in a neighborhood of the test element. Once partitioned, for the training set the *k* nearest data elements are encoded into Bloch vectors and used to define the quantum centroid of each class.

The local quantum-inspired classifier considered, for reasonable values of the hyperparameters, was found to be a method with performance comparable to classical algorithms for multi-class classification. We performed some experiments using benchmark datasets and found that local quantum-inspired classifiers were even more accurate than SVM with different kernels, a random forest, a naive Bayes classifier and the logistic regression classification algorithm.

The present proposal offers a family of classifiers. In fact, several strategies to impose a notion of locality over a training set, and several procedures of quantum state discrimination, can be applied. Both the local approach to classification and the quantum-inspired data encoding/processing deserve further investigation to clarify the impact of these ideas on machine learning, but the results achieved clearly indicate that both approaches to machine learning are promising.

## Figures and Tables

**Table 1 entropy-25-00404-t001:** Classification comparison, in terms of test average accuracy for 10 runs, for benchmark datasets.

Dataset/Method	Euclidean	Manhattan	Chessboard	Canberra	Bray–Curtis	Bures	Nearest Mean
analcatdata_aids	0.6	0.62	0.53	0.55	0.51	0.46	0.46
analcatdata_asbestos	0.794	0.794	0.794	0.782	0.794	0.794	0.8
analcatdata_bankruptcy	0.87	0.87	0.83	0.85	0.88	0.87	0.87
analcatdata_boxing1	0.725	0.721	0.696	0.688	0.721	0.688	0.646
analcatdata_cyyoung9302	0.811	0.794	0.844	0.789	0.811	0.817	0.833
analcatdata_dmft	0.201	0.199	0.198	0.191	0.202	0.209	0.18
analcatdata_happiness	0.475	0.458	0.383	0.433	0.475	0.45	0.325
analcatdata_japansolvent	0.79	0.82	0.8	0.74	0.81	0.77	0.74
analcatdata_lawsuit	0.979	0.974	0.981	0.968	0.974	0.972	0.976
appendicitis	0.866	0.811	0.867	0.876	0.886	0.838	0.857
biomed	0.917	0.91	0.9	0.898	0.91	0.91	0.91
breast_cancer	0.698	0.707	0.696	0.674	0.695	0.691	0.704
iris	0.943	0.957	0.95	0.947	0.947	0.943	0.937
labor	0.927	0.927	0.836	0.909	0.936	0.936	0.936
new_thyroid	0.974	0.981	0.96	0.956	0.977	0.97	0.984
phoneme	0.897	0.897	0.895	0.893	0.897	0.898	0.903
prnn_fglass	0.746	0.693	0.7	0.685	0.671	0.72	0.724
prnn_synth	0.874	0.864	0.896	0.856	0.878	0.894	0.9
tae	0.58	0.583	0.573	0.543	0.577	0.58	0.56
wine_recognition	0.989	0.983	0.975	0.967	0.972	0.981	0.981
**Dataset/Method**	**Linear**	**Radial Basis Function**	**Polynomial**	**Sigmoid**	**Random Forest**	**Naive Bayes**	**Logistic Regression**
analcatdata_aids	0.45	0.45	0.49	0.45	0.42	0.49	0.56
analcatdata_asbestos	0.7	0.724	0.694	0.735	0.694	0.735	0.724
analcatdata_bankruptcy	0.79	0.79	0.81	0.7	0.78	0.77	0.81
analcatdata_boxing1	0.621	0.654	0.65	0.629	0.713	0.658	0.654
analcatdata_cyyoung9302	0.778	0.756	0.767	0.7	0.778	0.75	0.817
analcatdata_dmft	0.175	0.175	0.17	0.168	0.18	0.182	0.191
analcatdata_happiness	0.35	0.3	0.325	0.283	0.358	0.35	0.442
analcatdata_japansolvent	0.77	0.77	0.76	0.72	0.75	0.8	0.75
analcatdata_lawsuit	0.955	0.953	0.955	0.904	0.955	0.945	0.964
appendicitis	0.852	0.857	0.842	0.743	0.867	0.848	0.867
biomed	0.857	0.857	0.864	0.838	0.857	0.881	0.857
breast_cancer	0.656	0.656	0.668	0.651	0.675	0.672	0.66
iris	0.94	0.927	0.93	0.87	0.933	0.923	0.937
labor	0.782	0.854	0.827	0.636	0.827	0.855	0.845
new_thyroid	0.963	0.958	0.947	0.958	0.949	0.947	0.96
phoneme	0.75	0.821	0.829	0.746	0.883	0.766	0.745
prnn_fglass	0.541	0.51	0.559	0.483	0.666	0.629	0.568
prnn_synth	0.85	0.85	0.844	0.848	0.846	0.844	0.86
tae	0.446	0.403	0.43	0.273	0.45	0.477	0.457
wine_recognition	0.978	0.978	0.978	0.947	0.975	0.975	0.983

**Table 2 entropy-25-00404-t002:** Classification comparison, in terms of test average accuracy for 10 runs, for benchmark datasets. For quantum-inspired methods we show the average execution time in seconds on an Intel i7-9750H CPU @ 2.60GHz 6-core with 32 GB RAM, and for classical methods the training time and evaluation time per test set element.

Dataset	Method	Accuracy	Execution time	Method	Accuracy	Training time	Evaluation Time
analcatdata_aids	Manhattan	0.62	0.013	LogisticRegression	0.56	0.942	0.0018
analcatdata_asbestos	NearestMean	0.8	0.018	Sigmoid	0.735	1.244	0.0015
analcatdata_bankruptcy	BrayCurtis	0.88	0.019	Polynomial	0.81	0.893	0.0015
analcatdata_boxing1	Euclidean	0.725	0.021	RandomForest	0.713	0.351	0.0054
analcatdata_cyyoung9302	Chessboard	0.844	0.074	LogisticRegression	0.817	1.201	0.0014
analcatdata_dmft	Bures	0.209	0.013	LogisticRegression	0.191	3.103	0.0018
analcatdata_happiness	Euclidean	0.475	0.021	LogisticRegression	0.442	1.416	0.0024
analcatdata_japansolvent	Manhattan	0.82	0.06	NaiveBayes	0.8	0.636	0.0045
analcatdata_lawsuit	Chessboard	0.981	0.074	LogisticRegression	0.964	2.243	0.0022
appendicitis	BrayCurtis	0.886	0.094	RandomForest	0.867	0.657	0.0087
biomed	Euclidean	0.917	0.194	NaiveBayes	0.881	0.682	0.0045
breast_cancer	Manhattan	0.707	0.339	RandomForest	0.675	0.832	0.009
iris	Manhattan	0.957	0.029	Linear	0.94	0.904	0.0015
labor	BrayCurtis	0.936	0.013	NaiveBayes	0.855	0.318	0.0038
new_thyroid	NearestMean	0.984	0.099	Linear	0.963	1.452	0.0015
phoneme	NearestMean	0.903	39.204	RandomForest	0.883	2.157	0.0093
prnn_fglass	Euclidean	0.746	0.378	RandomForest	0.666	1.646	0.01
prnn_synth	NearestMean	0.9	0.12	LogisticRegression	0.86	2.323	0.0024
tae	Manhattan	0.583	0.148	NaiveBayes	0.477	3.801	0.0065
wine_recognition	Euclidean	0.989	0.394	LogisticRegression	0.983	2.426	0.003

**Table 3 entropy-25-00404-t003:** kNNSVM comparison, in terms of test average accuracy for 10 runs, for benchmark multi-class datasets.

Dataset/Method	1NNSVM	2NNSVM	3NNSVM	4NNSVM	5NNSVM	6NNSVM	7NNSVM	8NNSVM	9NNSVM
analcatdata_dmft	0.186	0.198	0.198	0.15	0.177	0.187	0.187	0.187	0.196
analcatdata_happiness	0.242	0.317	0.325	0.375	0.383	0.383	0.375	0.375	0.35
iris	0.937	0.943	0.943	0.95	0.943	0.957	0.937	0.95	0.96
new_thyroid	0.974	0.965	0.958	0.96	0.958	0.965	0.949	0.951	0.96
prnn_fglass	0.72	0.71	0.658	0.678	0.683	0.678	0.666	0.673	0.676
tae	0.57	0.593	0.557	0.55	0.553	0.543	0.53	0.53	0.493
wine_recognition	0.986	0.981	0.986	0.989	0.989	0.989	0.983	0.983	0.986
**Dataset/Method**	**10NNSVM**	**11NNSVM**	**12NNSVM**	**13NNSVM**	**14NNSVM**	**15NNSVM**	**16NNSVM**	**17NNSVM**	**18NNSVM**
analcatdata_dmft	0.197	0.196	0.2	0.202	0.193	0.185	0.197	0.204	0.204
analcatdata_happiness	0.358	0.4	0.317	0.408	0.433	0.425	0.35	0.375	0.4
iris	0.95	0.95	0.95	0.957	0.95	0.953	0.943	0.937	0.943
new_thyroid	0.951	0.937	0.963	0.944	0.944	0.949	0.96	0.953	0.955
prnn_fglass	0.676	0.641	0.685	0.659	0.663	0.654	0.693	0.678	0.685
tae	0.49	0.433	0.52	0.48	0.43	0.427	0.48	0.467	0.487
wine_recognition	0.983	0.972	0.983	0.972	0.964	0.967	0.989	0.969	0.975

## Data Availability

The code is also available at the following repository: https://github.com/leporini/classification (accessed on 21 February 2023). Penn Machine Learning Benchmarks https://github.com/EpistasisLab/pmlb (accessed on 21 February 2023).

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
