# Peer review of "Quantum-Inspired Applications for Classification Problems"

_entropy, 2023, doi:10.3390/e25030404_

Round 1

Reviewer 1 Report

The authors present a variant of previous techniques designed to deal with the classification problem based on quantum state discrimination. The novelty of the current approach relies in that a local preprocessing of the information is carried out, based on considering K closest neighbors. Different variants with alternative measures are considered, and a numerical comparison with other methods is carried out, showing favorable results for the algorithms proposed.

The manuscript is sound and the results are promising. I would be happy to recommend this manuscript for publication after the following points are taken care of.

(1) When discussing the techniques based on the pretty good measurements approach, the authors should mention the following recent publication:

R. Giuntini et al, Quantum-inspired algorithm for direct multi-class classification,

Applied Soft Computing, Volume 134, 2023, 109956.

(2) The numerical results are presented in section 4. But the complexity of the proposed algorithms is not discussed. This is a crucial point, given that the real usefulness of these techniques relies on the possibility of implementing them in an efficient way. For example, the computation times and the hardware used should be communicated (in comparison with other non-quantum-like techniques). In particular: how much harder is to implement the “local” k-nearest neighbors variant with regard to the standard “global” variant?

(3) Are the algorithms presented here thought of as running on a purely classical hardware? Or do the authors think of implementing them on an actual quantum computer? This should be explicitly clarified in the manuscript, given that many researchers are not familiar with quantum-inspired algorithms.

Author Response

We thank the referees for their comments and suggestions. We added the article citation and a table with computation times and the characteristics of the hardware, as requested. Octree and KDtree construction can be done in the preprocessing phase efficiently with Mathematica's Nearest function. These experimental results are promising and show that the methods are efficient when run on classical computers. We will study in a future work how to apply the local methods also in implementations on quantum computers and analyze the complexity of such algorithms.

Reviewer 2 Report

The authors implemented some modifications of the classification algorithms based on quantum state discrimination. However, it is absolutely unclear from the manuscript why it is something that deserves the attention of the community, i.e. the advantages of this modification aren't discussed. Unless this information is provided, I do not see reasons for publishing it. The link to the code from the manuscript doesn't work, but I still managed to find the code on GitHub. The code is just a bunch of Mathematica files that do not have any discription, any comments, so I do not think that anyone would ever be able to use it (or to check it, for instance, as it was in my case).  There are infinitely many ways of modifying countless ML techniques, so one has to be very clear about why this particular modification deserves to be published. Even the authors say "Both the local approach to classification and the quantum-inspired data encoding/processing deserve further investigation to clarify the impact of these ideas on machine learning.". 

To address this concern on the advantages of the algorithm compared to the known, I suppose the authors have to give an example of solving some useful problem with a clear manifestation of the advantage of using their modification of the algorithm. For example, one may take this paper as a guideline [https://doi.org/10.1103/PhysRevB.99.041108], where the authors clearly proved that the use of the machine learning technique "self-organized maps" leads to huge savings of time when calculating the critical temperature compared to the standard methods, thus proving its usefulness. Unless the authors convince me with a use-case that their modification gives a significant advantage as compared to the known techniques and unless they turn a bunch of Mathematica files without any description and commets to a product one can use (or at least one can check), I can not give my positive recommendation.

Author Response

We thank the referees for helpful comments and suggestions. We published the commented Mathematica code on https://github.com/leporini/classification/tree/main/analcatdata_dmft with the dataset analcatdata_dmft  (797 observations, 4 features, 6 classes) taken from https://epistasislab.github.io/pmlb/ as requested. We have modified the introduction to highlight the advantages of quantum-inspired methods with a local approach. First of all, we perform multi-class classification directly, without using binary classifiers, based on Helstrom discrimination. As shown in the experimental results, local approach improves the accuracy in classification in support vector machines as in quantum-inspired methods, where k is equal to the number of classes (few training data). It has the potential to offer comparable performance using less complexity. The results achieved clearly indicate that both approaches to machine learning are promising.

Round 2

Reviewer 1 Report

The problem of the complexity is not addressed in the revised version either. The authors promise to do it in a future work. Given that the authors are not willing to follow my suggestion, my work is done here. I leave the decision in the hands of the Editor.

Author Response

We have added a concise discussion of complexity.

The first algorithm uses a k-d tree in the training set, while the other two use a k-d tree in the corresponding Bloch vector space. The time complexity to construct the k-d tree is usually O(d n log n), where n is the cardinality of the training set and d the length of each vector, while the space complexity is O(d n). The query to find the k nearest neighbors takes O(k log n). The time complexity of PGM is O(c d^3) and is O(d m) for the classification of the m elements of the test set with c classes.

Our algorithm is efficient in the presence of multiple copies because it remove nulls and duplicates. In particular, we consider only 20 values instead of 81 matrix elements of $\rho_{(x_1,x_2)}\otimes \rho_{(x_1,x_2)}$, 51 values instead of 729 for $\rho_{(x_1,x_2)}\otimes \rho_{(x_1,x_2)}\otimes \rho_{(x_1,x_2)}$ and so on. In a future paper, we will analyze in detail the complexity of such algorithms in the average case and in the worst case. For instance, one can construct the ball tree for clustered data instead of k-d tree and consider different search techniques.

Reviewer 2 Report

the authors almost entirely ignored the previous report, I do not recommend for publication, see the previous report

Author Response

The authors implemented some modifications of the classification algorithms based on quantum state discrimination. However, it is absolutely unclear from the manuscript why it is something that deserves the attention of the community, i.e. the advantages of this modification aren't discussed. Unless this information is provided, I do not see reasons for publishing it.

We added some considerations shared with other authors dealing with quantum inspired methods in the literature to motivate community attention.

Quantum-inspired machine learning has revealed how relevant benefits for machine learning problems can be obtained using the quantum information theory even without employing quantum computers illustrating the potential advantages in solving several problems.
Giuntini, R.; Holik, F.; Park, D.K.; Freytes, H.; Blank, C.; Sergioli, G.
Quantum-inspired algorithm for direct multi-class classification. Applied Soft Computing 2023, 134, 109956. https://doi.org/10.1016/j.asoc.2022.109956
PGM discriminates not only orthogonal states but also non-orthogonal states with a non-zero success probability. Thefore, quantum theory provides a richer data representation. Moreover, our PGM within algorithms is more efficient than the one proposed by these authors in the case of multiple preparations in the same state because it removes duplicates and null values in encoding.
Quantum inspired methods are used in applications that solve industry-relevant problems related to finance, optimization and chemistry \cite{MROR19,MKSFLLO22,MLO22,CCF21,AALRV21}.

The link to the code from the manuscript doesn't work, but I still managed to find the code on GitHub. The code is just a bunch of Mathematica files that do not have any description, any comments, so I do not think that anyone would ever be able to use it (or to check it, for instance, as it was in my case). 

We added a short description in github. In the analcatdata_dmft folder
https://github.com/leporini/classification/tree/main/analcatdata_dmft
there is a commented example with the dataset downloaded from https://epistasislab.github.io/pmlb
Just run the code in the same folder.
Certainly, more code development work is required to include the classifier in Mathematica functions (or in scikit package with Python).

There are infinitely many ways of modifying countless ML techniques, so one has to be very clear about why this particular modification deserves to be published. Even the authors say "Both the local approach to classification and the quantum-inspired data encoding/processing deserve further investigation to clarify the impact of these ideas on machine learning".  To address this concern on the advantages of the algorithm compared to the known, I suppose the authors have to give an example of solving some useful problem with a clear manifestation of the advantage of using their modification of the algorithm. For example, one may take this paper as a guideline [https://doi.org/10.1103/PhysRevB.99.041108], where the authors clearly proved that the use of the machine learning technique "self-organized maps" leads to huge savings of time when calculating the critical temperature compared to the standard methods, thus proving its usefulness. Unless the authors convince me with a use-case that their modification gives a significant advantage as compared to the known techniques and unless they turn a bunch of Mathematica files without any description and commets to a product one can use (or at least one can check), I can not give my positive recommendation.

First of all, we perform multi-class classification directly, without using binary classifiers, based on Helstrom discrimination. As shown in the experimental results, local approach improves the accuracy in classification in support vector machines as in quantum-inspired methods, where k is equal to the number of classes. It has the potential to offer comparable performance using less complexity. We have also added some complexity comments. The results achieved clearly indicate that both approaches to machine learning are promising.

Round 3

Reviewer 2 Report

none